# Flavonoid–Phenolic Acid Hybrids Are Potent Inhibitors of Ferroptosis via Attenuation of Mitochondrial Impairment

**DOI:** 10.3390/antiox13010044

**Published:** 2023-12-26

**Authors:** Madeline Günther, Samentha Dabare, Jennifer Fuchs, Sandra Gunesch, Julian Hofmann, Michael Decker, Carsten Culmsee

**Affiliations:** 1Institute of Pharmacology and Clinical Pharmacy, Philipps-University Marburg, Karl-von-Frisch-Str. 2, 35043 Marburg, Germany; madeline.guenther@pharmazie.uni-marburg.de; 2Marburg Center of Mind, Brain, and Behavior—CMBB, Hans-Meerwein-Str. 6, 35032 Marburg, Germany; 3Pharmaceutical and Medicinal Chemistry, Institute of Pharmacy and Food Chemistry, University of Würzburg, Am Hubland, 97074 Würzburg, Germanymichael.decker@uni-wuerzburg.de (M.D.)

**Keywords:** flavonoid–phenolic acid hybrids, ferroptosis, oxidative stress, mitochondria, metabolic effects

## Abstract

Cinnamic acid, ferulic acid, and the flavonoids quercetin and taxifolin (dihydroquercetin) are naturally occurring compounds found in plants. They are often referred to as polyphenols and are known, among others, for their pharmacological effects supporting health through the inhibition of aging processes and oxidative stress. To improve their bioavailability, pharmacological activities, and safety, the creation of novel flavonoid–phenolic acid hybrids is an area of active research. Previous work showed that such hybridization products of phenolic acids and flavonoids enhanced the resilience of neuronal cells against oxidative stress in vitro, and attenuated cognitive impairment in a mouse model of Alzheimer’s disease (AD) in vivo. Notably, the therapeutic effects of the hybrid compounds we obtained were more pronounced than the protective activities of the respective individual components. The underlying mechanisms mediated by the flavonoid–phenolic acid hybrids, however, remained unclear and may differ from the signaling pathways activated by the originating structures of the respective individual phenolic acids or flavonoids. In this study, we characterized the effects of four previously described potent flavonoid–phenolic acid hybrids in models of oxidative cell death through ferroptosis. Ferroptosis is a type of iron-dependent regulated cell death characterized by lipid peroxidation and mitochondrial ROS generation and has been linked to neurodegenerative conditions. In models of ferroptosis induced by erastin or RSL3, we analyzed mitochondrial (lipid) peroxidation, mitochondrial membrane integrity, and Ca^2+^ regulation. Our results demonstrate the strong protective effects of the hybrid compounds against ROS formation in the cytosol and mitochondria. Importantly, these protective effects against ferroptosis were not mediated by radical scavenging activities of the phenolic hybrid compounds but through inhibition of mitochondrial complex I activity and reduced mitochondrial respiration. Our data highlight the effects of flavonoid–phenolic acid hybrids on mitochondrial metabolism and further important mitochondrial parameters that collectively determine the health and functionality of mitochondria with a high impact on the integrity and survival of the neuronal cells.

## 1. Introduction

Aging is a natural and inevitable process with increased susceptibility to a range of age-associated diseases. Among these, neurodegenerative disorders such as dementia, Alzheimer’s (AD), and Parkinson’s disease (PD) represent an increasingly prevalent challenge for our aging societies [1]. Intensive research in this field has focused on the disturbed function of mitochondria, the energy supply centers of our cells. These organelles play pivotal roles in maintaining cellular function and survival. However, during aging processes, the integrity of mitochondria becomes increasingly compromised, leading to a cascade of detrimental effects [2,3]. A consequence is the overproduction of reactive oxygen species (ROS), which are capable of causing extensive damage to cellular components, including lipids, proteins, and DNA. Such oxidative stress, driven by ROS, further exacerbates mitochondrial dysfunction and contributes to the progression of neurodegeneration [4,5]. Moreover, a new concept in the research field of neurodegenerative diseases is the involvement of ferroptosis, a unique form of regulated cell death characterized by the release of free labile iron into the cytosol and the concomitant iron-dependent lipid peroxidation [6,7]. Emerging evidence suggests that mitochondrial ROS formation and damage to these organelles plays a key role in ferroptosis, in particular in neuronal cells [8,9]. Therefore, enhancing mitochondrial antioxidant capacity and maintaining mitochondrial homeostasis and integrity may provide significant therapeutic effects in aging processes and neurodegenerative disorders [10,11,12,13]. Indeed, targeting mitochondria in particular has become preferable for the therapeutics of neurodegeneration, since neurons are highly dependent on OXPHOS for their considerable energy demands and contain fewer antioxidants than other cells, which makes them highly susceptible to oxidative stress [10,14,15]. Furthermore, certain neurodegenerative disorders are linked to defects in the ETC complexes. Complex I and complex III [10] are heavily involved in ROS production over OXPHOS, consequently leading to hindered ATP formation and metabolic alterations. Current strategies to target mitochondria include the inhibition of the ETC complex, uncoupling of OXPHOS, controlling ROS and oxidative stress levels, or the regulation of Krebs cycle enzymes [10]. However, the targeted application of drugs and compounds toward mitochondria faces some difficulties, due to the permeability, high negative potential, and bilayer structure of mitochondria [10].

Thus, developing new molecules that can interfere with ferroptotic cell death pathways by preserving mitochondrial integrity is an emerging field of research. Natural polyphenolic compounds such as flavonoids and phenolic acids have demonstrated their potential to inhibit ferroptosis [16,17,18] and to alleviate the severity of neurodegeneration, among others, through anti-AD effects in several AD models owing to their antioxidative, anti-inflammatory, and anti-amyloidogenic properties [19,20,21,22]. 

Their antioxidant activity due to heterocyclic structures including double bonds in combination with a 4-oxo function is considered to be a key structural element [23,24] in reducing the severity of age-related diseases. In fact, many neurological disorders, type 2 diabetes, or vascular diseases can be caused or exacerbated by free radicals [4,5,6]. However, the bioavailability of flavonoids in the brain or bloodstream [25,26] appears far too low and thus insufficient for alleviating neurodegenerative diseases through their antioxidative capacities. This raised the question of alternative modes of action and activation of defined molecular pathways, such as the KEAP1–Nrf2 pathway, PPARγ activation, or inhibition of pro-inflammatory signaling [22,27,28]. 

Recently, a novel pharmacological class of flavonoid–phenolic acid hybrids was reported that showed improved antioxidant and anti-inflammatory activities [29]. Among a variety of such hybrid compounds, the 7-O-esters of taxifolin were presented as the most effective derivatives [29]. These promising compounds consist of cinnamic or ferulic acid, attached to the flavonoid taxifolin on position 7 of the benzene ring A. It turned out that the 7-O-esters UW-MD-189 (7-O-Cinnamoyltaxifolin) and UW-MD-190 (7-O-Feruloyltaxifolin) showed neuroprotective effects in mouse hippocampal-derived HT22 cells, which were more pronounced in comparison to the individual natural components taxifolin and cinnamic and ferulic acid alone or in combination. Similar results were obtained when testing these compounds in a model of LPS-induced inflammation in BV-2 microglia cells. Here, the 7-O-esters UW-MD-189 and UW-MD-190 reduced NO and IL-6 secretion and increased Nrf2 expression levels, indicating the anti-inflammatory properties of these hybrids. Further improvements in the structure–activity relationships of the flavonoid component led to the identification of taxifolin and quercetin as ideal coupled flavonoid molecules [30]. Recently, the additional improvement of the linker between flavonoid and cinnamic acid led to the exchange of the ester with an amide structure, resulting in improved hybrid stability. The cellular uptake of the hybrids was thereby increased, while a reduction in the effective working concentration was achieved. Intriguingly, the four resulting compounds UW-MD-189/190 (7-O-esters of taxifolin), UW-MD-457 (taxifolin cinnamic acid amide), and UW-MD-458 (quercetin cinnamic acid amide) showed improvements in short- and long-term memory when mice were injected with Aβ_25–35_ peptides.

To evaluate the potential mechanisms of action of the novel compounds, an alkylene-tagged 7-O-cinnamoyltaxifolin probe was applied for affinity pulldown and subsequent MS analysis [31]. The MS profile indicated that 7-O-cinnamoyltaxifolin (UW-MD-189) shows interaction with adenine nucleotide translocase 1 (ANT-1), an antiporter for the exchange of ADP from the cytoplasm to the mitochondrial matrix, and also with the sarco/endoplasmic reticulum Ca^2+^ ATPase (SERCA) [31]. ANT-1 and SERCA are both crucial proteins with distinct roles in mitochondrial calcium handling and the cellular stress response. Calcium released from the ER into the cytoplasm can be taken up by mitochondria, impacting mitochondrial function [32,33] and ANT-1 activity. As an integral membrane protein, ANT-1 is crucial for the formation of the mPTP complex and is thereby involved in the regulation of mitochondrial membrane permeability [34,35]. This in turn affects the electrochemical gradient and ATP synthesis. Overall, this represents a mechanistic link of novel compounds to mitochondrial homeostasis. 

Underlying these findings, studying the effects of the four flavonoid–phenolic acid hybrids UW-MD-457, 458, 189, and 190 on mitochondria, particularly in models of oxidative stress, offers valuable insights into cellular energy metabolism, oxidative damage, and the potential for therapeutic intervention. Our data support the understanding of how these compounds affect cellular function and provide a mechanistic basis for developing safe and effective treatments for diseases associated with mitochondrial dysfunction and oxidative stress.

## 2. Materials and Methods

### 2.1. Flavonoid–Phenolic Acid Hybrids

The directly connected hybrid molecules were synthesized as previously described in detail [29], i.e., cinnamic and ferulic acid, respectively, were activated using oxalyl chloride in anhydrous tetrahydrofuran (THF) with catalytic amounts of dimethylformamide for 1 h at room temperature. A subsequent reaction with taxifolin in dry THF (again for 1 h at room temperature) in the presence of triethylamine yielded both hybrid molecules at 43% (7-O-cinnamoyltaxifolin; UW-MD-189) and 27% (7-O-ferulyltaxifolin, UW_MD-190). This reaction sequence avoids the use of any protection group [29]. Spectral data were in accordance with the literature [29].

The amide hybrid molecules were synthesized as previously described in detail [30]: For this reaction sequence, a protection group strategy had to be applied with peracetylation of quercetin and taxifolin, respectively, and imidazole-mediated selective hydrolysis in position 7. The mono hydroxy compound reacted in a Williamson ether synthesis with a cinnamic acid amide bearing an iodine atom, followed by subsequent deprotection of the remaining acetyl groups, yielding the quercetin–cinnamic acid amide (UW-MD-458) and the taxifolin–cinnamic acid amide hybrids (UW-MD-457). Spectral data were in accordance with the literature [30].

### 2.2. Cell Culture

The mouse hippocampal neuronal cell line HT22 (Salk Institute, San Diego, CA, USA) was cultured in DMEM (DMEM High Glucose; Capricorn Scientific GmbH, Ebsdorfergrund, Germany) supplemented with 10% heat-inactivated fetal calf serum (Biochrom, Berlin, Germany), 100 U/mL penicillin, and 100 mg/mL streptomycin at 37 °C, 95% humidity, and 5% CO_2_ (HeracellTM 150; Thermo Fisher Scientific, Darmstadt, Germany).

### 2.3. Cell Viability

Cell viability was assessed via an MTT assay measuring the metabolic activity of HT22 cells. Therefore, cells were grown for 24 h in a 96-well plate format and treated as indicated. After 16 h of treatment, 2.5 mg/mL MTT stock solution was added to the culture medium, followed by further incubation of the plates for 60 min to allow the cells to convert MTT into purple formazan crystals through their metabolic activity. Afterward, the MTT solution was carefully aspirated and DMSO was added to each well to dissolve the formazan crystals. After rocking the plate gently, absorbance was measured at a wavelength of 570 nm, and background absorbance was corrected with a reference wavelength of 630 nm. Cell viability was calculated as a percentage by comparing the absorbance values of treated cells to the control cells. A dose–response curve and EC50 values were calculated using GraphPad Prism software 8.2.1 (GraphPad Software Inc., La Jolla, CA, USA) with data obtained from three to four independent experiments.

### 2.4. Cell Proliferation

To assess cell proliferation and monitor cellular behavior in real time, an xCELLigence system impedance measurement (Agilent, Waldbronn, Germany) was conducted. Before seeding the cells in a 96-well E-Plate, the instrument was calibrated according to the manufacturer’s instructions. Afterward, cells grew for 24 h until compound treatment was applied. Over time, the impedance changed according to the increase in cell number or cellular detachment from the plate. For data analysis, RTCA software 1.2 (Roche Diagnostics, Penzberg, Germany) normalized the proliferation curves of the treatment application to 1 before real-time growth curves and cell index values were generated.

### 2.5. Cell Death

Annexin V and PI staining is a valuable tool to study cell death and characterize the state of cells based on the integrity of their plasma membranes. HT22 cells were collected from 24-well plates by washing the cells with PBS, detaching them with trypsin, and collecting them in 1.5 mL tubes. After a washing step with PBS, cells were stained with Annexin-V-FITC (PromoKine Annexin V-FITC Detection Kit, Promokine, PK-CA577-K101-25/100/400) and propidium iodide (PI) for 10 min in the dark at room temperature. Immediately, cell death was detected using the Guava EasyCyte flow cytometer (Merck Millipore, Darmstadt, Germany), exciting the Annexin-V-FITC at 488 nm and detecting the signal with a 525 nm bandpass filter. The PI signal was excited at 488 nm and detected by a 690 bandpass filter. Data were obtained from at least 5000 cells per replicate. Annexin V-positive and PI-positive cells were quantified as dead cells by the Guava 3.1.1 Software (Merck Millipore, Darmstadt, Germany).

### 2.6. (Mitochondrial) Lipid Peroxidation

The assessment of cytosolic and mitochondrial lipid peroxidation allows for the quantitative analysis of oxidative stress in the cells. HT22 cells were treated as indicated and stained after 16 h of incubation with 0.5 µg/mL/well C11-BODIPY (Invitrogen, Karlsruhe, Germany) or 1 µg/mL MitoPerOx (Abcam, Cambridge, UK; GB) for 30 min in the cell culture incubator. Afterward, cells were washed to remove the unbound probe and collected in a microcentrifuge tube. Filters for signal detection were adjusted to 525/30 nm for excitation of the green fluorescence signal and to 690/50 nm for emission of the red fluorescence signal. BODIPY and MitoPerOx oxidation were measured from at least 5.000 cells per replicate per condition and analyzed by recording the shift from red to green fluorescence with the Guava EasyCyte flow cytometer (Merck Millipore, Darmstadt, Germany).

### 2.7. Mitochondrial Superoxides (O_2_^•−^)

Mitochondrial superoxides were detected by the use of MitoSOX Red (Invitrogen, Karlsruhe, Germany), a fluorescence probe specifically sensitive to superoxide radicals that accumulate within mitochondria. Cells of interest were harvested and washed with PBS to remove any residual culture medium or treatment agents. In a microcentrifuge tube, the cells were stained with 1 µg/mL MitoSOX Red for 30 min at 37 °C in the dark. To remove the unbound probe, cells were washed and resuspended with PBS. With the Guava flow cytometer (Merck Millipore, Darmstadt, Germany), MitoSOX Red was excited by a 488 nm laser and emitted red fluorescence at 690/50 nm upon oxidation by superoxide radicals.

### 2.8. Mitochondrial Membrane Potential (Δψm)

For the analysis of mitochondrial membrane potential, cells were stained with 0.4 mM TMRE (MitoPT ΔΨm Kit, ImmunoChemistry Technologies, Hamburg, Germany) for 30 min at 37 °C in the cell culture incubator. After harvesting the cells from the 24-well plates with trypsin and a subsequent washing step, cells were resuspended in an appropriate amount of PBS. TMRE fluorescence as an indicator for the integrity of the mitochondrial membrane potential was measured via FACS analysis with an excitation of 488 nm and emission at 690/50 nm.

### 2.9. Mitochondrial Ca^2+^

Specific evaluation of mitochondrial calcium levels was achieved by staining the cells with the selective dye Rhod-2 AM (rhodamine-2 acetoxymethyl ester; Thermo Fisher Scientific, Darmstadt, Germany). Before harvesting the cells with trypsin, they were loaded with Rhod-2 AM 1 µM working solution and incubated for 45 min at 37 °C with a serum-free medium. This allows Rhod-2 AM to enter cells, where it is cleaved by esterases to release the active Rhod-2 dye, which then targets the mitochondria and binds to calcium. After the incubation, cells were washed with PBS to remove excess Rhod-2 AM and unincorporated Rhod-2 dye. The fluorescence of Rhod-2-loaded mitochondria was assessed by FACS with excitation at 550 nm and emission at 580 nm. Higher fluorescence signals indicated an uptake of calcium into the cells.

### 2.10. Cytosolic Iron

To measure free labile Fe^2+^ in cells, the fluorescent heavy metal indicator Phen Green SK diacetate (Cat#25393, Cayman Chemical, Ann Arbor, MI, USA) with a stock concentration of 1500 µM was used as an indicator. Cells were harvested after the indicated treatment and washed once with PBS. Afterward, cells were stained with a final concentration of 5 µM Phen Green SK for 15 min and assessed by FACS analysis with an excitation wavelength of 488 nm and an emission wavelength of 525/30 nm subsequently. Phen Green SK diacetate fluorescence was quenched upon free labile iron exposure. 

### 2.11. Protein Analysis

In order to harvest proteins as indicated, cells were washed with PBS and lysed in 0.25 M Mannitol, 0.05 M Tris, 1 M EDTA, 1 M EGTA, 1 mM DTT, and 1% Triton-X containing lysis buffer enriched with Complete Mini Protease Inhibitor Cocktail and PhosSTOP (Roche Diagnostics, Penzberg, Germany). After centrifuging the samples for 10 minutes at 10,000× *g* and 4 °C, the supernatant was collected and protein content was analyzed using the Pierce BCA Protein Assay Kit (Perbio Science, Bonn, Germany). Using SDS-PAGE gel electrophoresis (10% polyacrylamide gel), proteins were separated to their size accordingly and transferred on a PVDF membrane for 1.5 h at 325 mA. To block non-specific protein binding on the membrane, 5% milk in TBST was used. Primary antibodies for GPX4 (1:1000, Abcam, Cambridge, UK), MCU (1:1000, Cell Signaling, Leiden, The Netherlands), and α-tubulin (1:10,000, Sigma Aldrich, Taufkirchen, Germany) were incubated overnight at 4 °C. Corresponding HRP-labeled secondary antibodies were incubated for 1 h at room temperature before the respective proteins were detected using the Chemidoc Imaging System (Bio-Rad, Munich, Germany). With Image Lab 4.0.1 Software (Bio-Rad, Munich, Germany), proteins were quantified and background-corrected intensities of the Western blot bands were determined, followed by intrinsic normalization to the respective loading control. To ensure comparability between different experiments, band intensities were compared to the respective control set as 100%.

### 2.12. Seahorse Bioenergetic Flux Analysis

The cellular bioenergetic parameters, mitochondrial respiration, and glycolysis were assessed in real time using the Seahorse XF96 Analyzer (Agilent Technologies, Waldbronn, Germany). HT22 cells were seeded in XF96-well microplates (7000 cells/well, Seahorse Bioscience) and incubated for 24 h. After the indicated treatment, the cell growth medium was replaced by the seahorse assay medium (4.5 g/L glucose, 2 mM glutamine, 1 mM pyruvate, pH 7.35 with NaOH), and cells were maintained for 1 h in a carbon dioxide (CO_2_)-free incubator. To ensure accurate measurement, the XF Analyzer was calibrated with a seahorse calibration solution before running the assay. Afterward, three baseline measurements were conducted, followed by injections of oligomycin (3 µM final concentration), FCCP (0.5 µM final concentration), rotenone/antimycin A (0.1 µM/1 µM), and 2-DG (246 mM). The cellular content of each well was analyzed by the BCA method immediately after the measurement. In accordance with protein content per well, data were normalized and assessed using Seahorse Wave 2.6.1 software (Agilent Technologies, Waldbronn, Germany). 

### 2.13. Radical Scavenging Activities

Antioxidant activity of the compounds was determined by DPPH (2,2-diphenyl-1-picrylhydrazyl, Cat#14805-50, Cayman Chemical, Ann Arbor, MI, USA) assay, based on the reduction of the purple-colored DPPH radical to a yellow-colored diphenylpicrylhydrazine. After preparation of DPPH solution in 90% ethanol and a final concentration of 150 µM, UW-MD compounds were added and incubated for 1 h at room temperature. Absorbance was assessed at 517 nm by a SPARK 20M plate reader (Tecan, Crailsheim, Germany), and the DPPH-scavenging effect was calculated via (A0 − AX)/(A0) × 100; A0: absorbance 90% ethanol, AX: absorbance of individual samples. 

### 2.14. ATP Level

An ATP (adenosine triphosphate) luminescence assay (Lonza, Verviers, Belgium) was used to quantify the amount of ATP in the HT22 cells. In accordance with the manufacturer’s instructions, a lysis buffer was prepared and added to the cells to ensure appropriate homogenization and the release of ATP into the supernatant. Transferring the supernatant into a white-walled 96-well plate, the luciferase enzyme and luciferin substrate were combined to create the working reagent, which was immediately added to the supernatant of the cells. The luciferase enzyme reacted with the ATP derived from the samples and standards to produce luminescence. Luminescence in relative light units (RLU) was detected with a Fluostar OPTIMA plate reader (BMG Labtech, Offenbach, Germany).

### 2.15. Mitochondrial Morphology

To examine mitochondrial morphology, MitoTracker^®^ DeepRed FM (Invitrogen, Karlsruhe, Germany) was employed to visualize active mitochondria through far-red fluorescence. Cells were incubated with 200 nM Mito Tracker, diluted in DMEM, for 15–30 min at 37 °C. Subsequently, an epifluorescence microscope (DMI6000B, Leica, Wetzlar, Germany) equipped with a 100×/1.4 NA oil immersion objective was utilized for visualization. Excitation occurred at a wavelength of 633 nm, and emitted light was detected at 670 nm. Mitochondrial morphology changes were quantified by the FIJI Mitochondrial Analyzer plugin. For background correction, the 2D threshold was adjusted to a block size of 5 microns, and an assessment of mitochondrial branches, branch junctions, total branch length, and mean area was performed for every cell individually.

## 3. Results

### 3.1. Flavonoid–Cinnamic and –Ferulic Acid Hybrids Protect HT22 Cells from Oxidative Stress-Induced Cell Death

First, we studied the effects of the flavonoid–cinnamic and –ferulic acid hybrids (Figure 1A,B) on oxidative dysregulation in mouse hippocampal HT22 cells. RSL3, erastin, and glutamate were used to induce ferroptosis through reduced GPX4 activity and enhanced lipid peroxidation [36,37]. High millimolar concentrations of extracellular glutamate or micromolar concentrations of erastin prevent cystine import through xCT, thereby resulting in decreased GSH levels, whereas RSL3 directly inhibits GPX4 activity [38], concomitantly increasing lipid peroxide production and ROS accumulation in the cells. Treatment of HT22 cells with RSL3, erastin, or glutamate resulted in an >85% reduction of metabolic activity, as determined by the MTT assay (Figure 1C–J). The flavonoid–cinnamic and –ferulic acid hybrids prevented ferroptosis-induced cell death at 3.12–12.5 µM (UW-MD-457), 0.78–12.5 µM (UW-MD-458), and 10–20 µM (UW-MD-189 and UW-MD-190) in a concentration-dependent manner (Figure 1C–J). This was in line with the EC_50_ values obtained from the compound concentration resulting in 50% protection against the corresponding ferroptotic activator (Table 1).

Among them, the compound UW-MD-458 was the most potent structure against the oxidative insult with an effective working concentration of 0.78 µM. Furthermore, the EC_50_ values of the four cinnamic/ferulic acid hybrids obtained from the different paradigms of ferroptosis did not differ much irrespective of whether cell death was induced by RSL3, erastin, or glutamate. Here, no significant differences were found for the protective effects of 189 and 190, regardless of the ferroptotic model. The compound UW-MD-458 consistently showed more potent effects against ferroptosis compared to UW-MD-457, but this was also associated with increased toxicity in a concentration-dependent manner (Figure 1C–F and Figure 2C,E). These and the previously obtained effects of the flavonoid–cinnamic/ferulic acid hybrids were confirmed in real-time proliferation measurements (Figure 2B,D,F,H,J and Appendix A) and by cell death assays using Annexin V and propidium iodide (Annexin V/PI) after erastin and RSL3 challenges for 16 h (Figure 2C,E,G,I). 

To assess whether these effects of the four hybrids were specific for the prevention of oxidative stress or whether they also affected other paradigms of cell death, we triggered apoptosis by staurosporine, a broad-spectrum inhibitor of protein kinases, established for the induction of caspase-3-dependent cell death [39]. Apoptotic stimuli can result in cytosolic Ca^2+^ influx, resulting in pro-apoptotic Bax, Bad, and p53 activation. Interestingly, these factors act on mitochondria to induce Ca^2+^ influx, oxidative stress, or the opening of the permeability transition pore (PTP) and finally lead to the release of cytochrome c following additional caspase-9/3 activation and execution of the apoptotic cell death process [33,40]. The pharmacological application of the antioxidants trolox and the mitochondrial radical scavenger MitoQ was not able to interfere with this model of caspase-dependent apoptosis (Appendix A). Similarly, the previously described complex I inhibitor of the electron transport chain (ETC), phenformin, did not affect the apoptotic cell death mechanism, in contrast to the caspase inhibitor QvD-OPh, which mitigated staurosporine-mediated apoptosis. These results indicate that neither antioxidant activities nor mitochondrial protection affected the progression of caspase-dependent apoptosis. There was also no effect of any of the four flavonoid–phenolic acid hybrids when inducing apoptosis by staurosporine; only 457 exerted mild protective effects (Appendix A). There is evidence suggesting that endoplasmatic reticulum (ER) stress may contribute to the initiation of ferroptosis. ER stress can increase the accumulation of reactive oxygen species (ROS) when activating the unfolded protein response (UPR) and, in turn, may further contribute to lipid peroxidation, which is a key feature of ferroptosis. However, in paradigms of tunicamycin-induced ER stress, the flavonoid–phenolic acid hybrids exhibited little to no capacity to prevent the accumulation of misfolded proteins, concomitantly leading to cell death (Appendix A). Interestingly, the antioxidants trolox and MitoQ, and also the pan-caspase inhibitor QvD-OPh, failed to mediate any protective effects in this death signaling pathway (Appendix A). Together, these findings suggest that all compounds selectively mitigate ferroptotic cell death induced by oxidative dysregulation while providing limited or no protection against ER stress or caspase-dependent apoptotic cell death. 

### 3.2. Flavonoid Amide and Ester Hybrids Suppress Intracellular ROS and Mitochondrial (Super)Oxide Production

Ferroptosis is defined by three essential hallmarks: the loss of lipid peroxide repair capacity, an increase in redox-active iron, and oxidation of polyunsaturated fatty acid (PUFA)-containing phospholipids [41] leading to the formation and accumulation of ROS. Therefore, we then studied the effect of the flavonoid amide and ester hybrids on intracellular and mitochondrial lipid peroxidation using the fluorescent probes C11-Bodipy and MitoPerOx, respectively. The HT22 cells were treated with erastin or RSL3 to induce ferroptosis, and flavonoid–phenolic acid hybrids were applied simultaneously. Afterwards, in the erastin and RSL3 treatment conditions, intense signals for cytosolic (Figure 3) and mitochondrial lipid peroxidation (Figure 4) were observed compared to the controls. However, these signals were strongly reduced in the presence of the flavonoid hybrids at their respective EC_50_ values (Figure 3D–G and Figure 4A,C,E,G). 

Investigation of the antioxidative capacities of the flavonoid hybrids by the DPPH assay (Figure 3B) showed that this protection against lipid peroxidation seems to be only partly mediated by their radical scavenging properties. In comparison to trolox, a vitamin E analog with potent antioxidative capacity, the hybrids only revealed moderate antioxidative properties, not reaching the radical scavenging effects of 50 µM trolox, which, in turn, were needed to mediate significant protection against ferroptosis (Figure 3C).

In order to study different aspects of mitochondrial ROS production, we used a MitoSOX probe to detect mitochondrial superoxide (O_2_^•−^) formation, the chief ROS in mitochondria, which are mainly derived from an electron leak in ETC complexes I and III. Most of these superoxides get converted into hydrogen peroxide (H_2_O_2_) either by natural dismutation or by SOD [10]; these were previously detected by MitoPerOx and Bodipy dye. When measuring MitoSOX signals after 16 h of RSL3 or erastin challenge, we observed a strong increase in mitochondrial superoxide formation (Figure 4B,D,F,H). These were again reliably prevented by co-treating the cells with the different UW-MD derivatives, indicating that the novel compounds were able to interfere with the mitochondrial mechanisms early in the ferroptotic cell death process. Mitochondrial superoxide formation is considered a key amplifying mechanism in the formation of highly reactive hydroxyl radicals [42], which are able to initiate lipid peroxidation cascades in membranes and therefore represent a major cause of cellular damage in ferroptosis.

### 3.3. Flavonoid–Cinnamic and –Ferulic Acid Hybrids Both Preserve Mitochondrial Membrane Integrity and Maintain Mitochondrial Calcium Levels

Mitochondrial membrane potential (MMP) and intracellular Ca^2+^ play a critical role in mitochondrial homeostasis and function [43]. Evidence indicates that the loss of MMP coincides with the opening of the mPTP, leading to the release of cytochrome c into the cytosol, which in turn triggers other downstream events in the ferroptotic cascade [44]. In this study, MMP was detected by TMRE fluorescence staining. However, erastin and RSL3 provoked the depolarization of the negatively charged mitochondria visible by the loss of red fluorescence intensity, due to the lost ability to sequester the positively charged TMRE. The flavonoid–cinnamic esters and amides (UW-MD-457/458 and UW-MD-189), as well as the flavonoid–ferulic-acid ester of taxifolin (UW-MD-190), reliably preserved mitochondrial membrane potential after the oxidative insult (Figure 5J–M). The protective effects of the cinnamic acid hybrids UW-MD-457 and UW-MD-458, thus also of the 7-O-esters of UW-MD-189 and UW-MD-190, were mediated in a concentration-dependent manner. Concentrations of UW-MD-458 higher than 6.25 µM reduced MMP-independent of erastin- or RSL3-mediated damage, whereas UW-MD-457, -189, and -190 alone did not impair mitochondrial membrane integrity. 

A healthy ΔΨm is important for maintaining mitochondrial and cellular health [45]. Previously, it was shown that a disbalance in ΔΨm can drive calcium uptake into mitochondria, which in turn modulates various pathways involved in ferroptosis, including those related to iron metabolism, lipid metabolism, and anti-oxidative defense [45]. Further, calcium can influence the activation of enzymes such as lipoxygenases that promote lipid peroxidation and ROS production. Therefore, mitochondrial calcium signaling pathways represent a potential target for the pronounced effects of the flavonoid hybrids against ferroptosis at the level of mitochondria. To investigate the effects of the four compounds on mitochondrial calcium signaling, we assessed the mitochondrial Ca^2+^ concentration by detecting the fluorescence intensity of the mitochondrial Ca^2+^ indicator Rhod-2 AM with a flow cytometer. The results showed that ferroptotic stress conditions caused by erastin or RSL3 significantly increased the mitochondrial Ca^2+^ concentration after only 10 h (Appendix A). However, UW-MD-189/190 and UW-MD-457/458 restored the ferroptosis-induced Ca^2+^ overload in mitochondria and maintained these levels in a concentration-dependent manner during the oxidative challenge for 16 h (Figure 5A–I), indicating that oxidative stress is closely connected to mitochondrial Ca^2+^ overload and also the ability of the discovered compounds to interfere in this signaling process. 

Notably, the hybrids did not alter MCU expression levels after 8 h as observed by Western blotting experiments, indicating that stabilization of mitochondrial Ca^2+^ levels is not predominantly mediated by regulating the mitochondrial calcium uniporter MCU (Figure 6E,F), but rather by blocking the calcium overload at an early stage independent of the MCU. Therefore, we investigated alterations in the antioxidative capacity by analyzing the expression levels of glutathione peroxidase GPX4, which plays a pivotal role in the neutralization of and defense against reactive oxygen species [46]. When investigating GPX4 protein levels after 8 h of incubation with the compounds and RSL3 in combination, it became evident that the flavonoid hybrids do not exert their beneficial effects by enhancing the glutathione-GPX4 antioxidative defense system, as GPX4 protein levels were not affected by the four compounds (Figure 6E,F). 

Elevated mitochondrial calcium levels can also influence (mitochondrial) iron homeostasis by the regulation of iron uptake, iron–sulfur cluster biogenesis, and the activity of iron-dependent enzymes. The mitochondrial calcium and iron crosstalk thus has implications for ROS formation and is critical evidence of ferroptosis. In order to analyze an altered iron concentration in the cytosol upon ferroptotic stress conditions, we labeled the HT22 cells with the fluorescent indicator Phen Green SK and analyzed cytosolic iron content after 8–10 h of RSL3 treatment with and without the flavonoid–phenolic acid compounds. Flow cytometry revealed an increase in the labile cytosolic iron pool after treatment with RSL3, apparent by the quenched proportion of Phen Green SK (Figure 6A,B), which was strongly reduced by the flavonoid hybrids (Figure 6C,D). All four UW-MD compounds mediated a significant reduction in the free cytosolic iron content already below their corresponding EC_50_ concentrations. Reducing the cellular labile iron availability by the flavonoid–phenolic acid analogs hence can contribute to their antioxidant effect by minimizing the production of ROS through the reaction of ferrous ions (Fe^2+^) with hydrogen peroxide H_2_O_2_ (Fenton reaction).

### 3.4. Flavonoid–Phenolic Acid Hybrids Reduce Mitochondrial Oxidative Metabolism

Ferroptotic cell death is characterized by the accumulation of reactive oxygen species leading to disturbed calcium and iron homeostasis. This is accompanied by impaired mitochondrial function. To assess changes in mitochondrial bioenergetics as an important parameter for mitochondrial function, we detected the mitochondrial oxygen consumption rate (OCR) after UW-MD treatment in neuronal HT22 cells (Figure 7). After 16 h of incubation, we found that basal respiration and maximal respiration were reduced in UW-MD-treated cells, suggesting strong effects of the flavonoid–phenolic acid hybrids in mitochondria. These effects were more pronounced for the flavonoid hybrids UW-MD-457 and UW-MD-458, which contain an amide bond linking the phenolic acids. Here, EC_50_ concentrations of these compounds mediated significant suppressing effects on the mitochondrial oxidative phosphorylation activity, whereas UW-MD-189 and UW-MD-190 meditated robust effects on mitochondrial function at concentrations starting from 10 µM. 

Distinct from oxidative phosphorylation, the cells were able to meet their energy demand through glycolysis. To assess the influence of the compounds on the alternative metabolic pathway, the extracellular acidification rate (ECAR), as an indicator for the glycolytic activity of the hippocampal cells, was assessed simultaneously by Seahorse fluxmeter measurements (Figure 8A,D,G,J). The metabolic flux analysis after 16 h of compound treatment revealed only slight effects of the hybrids on basal glycolytic activity; however, the compounds UW-MD-457 and UW-MD-458 had a strong effect on the glycolytic reserve of the neuronal cells. As indicated by the loss of glycolytic reserve, the cells were not able to further increase their glycolytic activity in response to elevated energy demands or cellular stress and already relied on glycolysis for energy provision even in the presence of oxygen (Figure 8B,E,H,K). Strikingly, ECAR regulation was more pronounced with the amide hybrids compared to the 7-O-esters UW-MD-189/190, since concentrations of their respective EC_50_ values already displayed significant effects on the glycolytic reserve. To investigate the consequences of these findings, we then assessed the ATP levels after UW-MD treatment under basal conditions and after induction of ferroptosis. ATP luciferase analyses were performed 10 h after the onset of treatments and revealed that ATP levels were not altered under basal culture conditions but were significantly preserved by all UW-MD compounds compared to the ferroptotic controls (Figure 8C,F,I,L). 

To further differentiate the mitochondrial effects of the four hybrid compounds, we assessed their activity on mitochondrial electron transport chain complexes directly by assessing changes in the OCR in permeabilized cells. This was achieved by selective permeabilization of the outer cellular membrane with saponin while preserving the mitochondrial organelle. Individual application of rotenone to the mitochondrial membrane decreased complex I activity, which was promoted by pyruvate, glutamine, and malate supplementation in the assay medium in advance. Basal OCR levels therefore allow for the functional assessment of complex I activity. As shown in Figure 9, UW-MD-457 and UW-MD-458 application displayed a significant downregulation of complex I activity. These effects became visible after only 3 h of hybrid compound incubation and, again, were more pronounced for the amide hybrids UW-MD-457 and UW-MD-458. 

However, when applying the complex II substrate succinate, no acute effects of any flavonoid hybrid compound were detected on succinate dehydrogenase (SDH), known as complex II of the electron transport chain (Appendix A). Similarly, complex IV regulation was assessed by the provision of a TMPD substrate (Kovac’s reagent), capable of donating electrons to cytochrome c and transferring them to molecular oxygen to generate water and produce ATP as a final step in the OXPHOS process. The Seahorse analysis after adding TMPD revealed that the compounds were unable to mediate significant effects on complex IV activity compared to the controls (Appendix A). Moreover, the long-term measurement in permeabilized neuronal HT22 cells over 9 h (Appendix A) showed that the mitochondrial effects of the compounds are primarily mediated via the regulation of complex I activity. These effects are particularly pronounced for compound UW-MD-457, as a strong decrease in OCR was detected after its acute injection, which can indeed be observed moderately for the other compounds. These results indicated the acute effects of the hybrid compounds on the electron transport chain, with strong implications for mitochondrial metabolism and energy supply.

### 3.5. Flavonoid–Phenolic Acid Analogs Preserve Mitochondrial Morphology

To rule out toxic effects of the compounds on mitochondria leading to mitochondrial damage and finally to the reduction in mitochondrial bioenergetics as detected, we next examined the mitochondrial morphology of the compound-treated HT22 cells using MitoTracker staining (Figure 10A). Investigation of the key parameters of the mitochondrial morphology (Figure 10B–E) revealed that RSL3-induced ferroptosis is accompanied by mitochondrial fragmentation, observed by the loss of elongated and interconnected mitochondrial structure, and further by a smaller and punctate rounded shape of the organelles [8]. Morphological features were quantified by the reduction in count and length of branches per mitochondria, the decrease in branch junctions, and the reduction in mitochondrial area [47]. The compounds mediated a rescue of these morphological aspects, despite challenging the cells with RSL3, implying that mitochondrial morphology and, as a consequence, mitochondrial functions were preserved in the presence of the hybrid compounds—not only under basal conditions but also under conditions of oxidative stress.

## 4. Discussion

In this study, we showed that the hybridization of the natural products cinnamic acid with taxifolin (UW-MD-457, UW-MD-189), cinnamic acid with quercetin (UW-MD-458), and ferulic acid with taxifolin (UW-MD-190) led to a new class of phenolic acid flavonoid hybrid compounds that reliably prevented cell death through mitochondrial protection in models of oxidative death. The observed neuroprotective activities in paradigms of ferroptosis were comparable among the novel hybrid compounds, in accordance with their respective EC_50_ concentration. Here, the obtained working concentrations were considerably lower than the effective concentrations of the corresponding individual natural products reported for models of oxidative stress [28,48]. 

Previous studies on structure–activity relationships declare that the 2,3 C-C double bond in the C-ring of the flavonoid structure mediates powerful antioxidative activities [49,50]. Our data show that the hybrid UW-MD-458 with the C-C double bond consistently mediated protection against ferroptotic cell death in three-fold lower concentrations in comparison to its analog UW-MD-457 without the double bond in the 2,3 position of the C-ring. In addition, it was shown that this C2-C3 double bond mediates increased stability against hydrogen bonds, responsible for the inactivation of the flavonoid molecules, further explaining the stronger action of UW-MD-458 in comparison to the other UW-MD derivatives. Similarly, the phenolic carboxylic acids used for hybridization of the novel compounds belong to the subclass of hydroxycinnamic acids, characterized by a phenol ring substituted with one or more hydroxyl (-OH) groups and a three-carbon propenoic acid side chain. This structure is well known for its antioxidant potential due to its ability to form a resonance-stabilized phenoxy radical [24]. Our results now demonstrate that a different substitution pattern of the phenol ring did not significantly affect the activity of the compounds in models of oxidative stress, as the compounds UW-MD-189 and UW-MD-190 showed similar protection against ferroptotic cell death. An outstanding improvement in the effectiveness of the compounds, however, was achieved by amidation instead of esterification of the acidic residue, resulting in the lower working concentrations of UW-MD-457 and UW-MD-458 compared to UW-MD-189 and UW-MD-190. Further reports suggest that this modification of the carboxylic acid group is necessary to overcome the blood–brain barrier in order to improve neurodegenerative diseases [51]. Together, these data demonstrate that the hybridization of flavonoids and phenolic acids represents a valuable improvement in the bioavailability of natural products, and this is also reflected at the cellular level. 

Strong antioxidative capacities of the flavonoids taxifolin and quercetin as well as the phenolic acids cinnamic and ferulic acid have been reported before [49]. Therefore, we used the DPPH assay to test whether these antioxidative properties were preserved during hybridization. Indeed, we found that all four hybrids mediated concentration-dependent radical scavenging activities. These actions were confirmed in the assessment of cytosolic and mitochondrial lipid peroxidation. The novel compounds abrogated the formation of ROS, again in a concentration-dependent manner. This is in line with the actions described for polyphenolic natural products, showing that their protection against lipid peroxidation is beneficial not only in models of AD but also in cardiovascular, pulmonary, and kidney diseases [28]. However, the ability of natural products, especially quercetin and taxifolin, in ferroptosis-associated disease raises the question of whether these beneficial effects are only mediated by the proposed antioxidative properties or whether additional signaling pathways are affected by these compounds. In line with this, antioxidants such as ferulic acid and analogs have been reported to provide additional signaling effects and are capable of mediating anti-inflammatory, antiangiogenic, or antihyperglycemic effects [48,52,53,54,55,56]. Further supporting this suggestion, we found in the DPPH assay that the compounds were not able to reach the same antioxidative capacity as trolox, an established vitamin E analog that mediates protective effects mainly through radical scavenging capacities. Thus, additional protective mechanisms are likely involved in the observed protective effects of the new hybrid compounds against oxidative cell death through ferroptosis. 

Besides the role of ROS in oxidative cell death, mitochondria are increasingly appreciated as an essential component of ferroptosis progression and impairment of their function seems to be highly involved in cellular demise [7,57]. In our study, it was therefore of great interest to analyze the effects of the novel hybrids on characteristic mitochondrial properties. How exactly flavonoids target mitochondria is still an open field in research and context dependent [58]. In particular, the flavonoid quercetin prevented ferroptosis by targeting mitochondria in renal and liver injury, and also in model systems of neural injury, but on the contrary, this natural product promoted an increase in cellular iron (Fe) content, initiating ROS overproduction and ferroptosis in several cancer cell lines [59]. However, our study clearly showed that the four compounds exhibited not only preserved mitochondrial morphology but also strong protection against mitochondria-derived ROS formation, which was shown to be an important component of ferroptosis progression in the model of cysteine-deprived oxidative cell death [4,5,7,57]. There are several mechanisms described to influence and contribute to the regulation of such harmful oxygen species.

Among them, maintaining proper iron homeostasis is essential for preventing oxidative stress and lipid peroxidation. It is still under debate whether flavonoids can also chelate transition metals, particularly free labile iron, which is highly involved in the generation of ROS through the Fenton reaction [6]. Quercetin is reported to regulate iron metabolism indirectly by inhibiting ferritin degradation or by hepcidin hyper-induction, thus contributing to a depressed iron pool [59,60]. Mitochondria are a major site of iron metabolism and are particularly affected by increased labile iron during the ferroptotic event. Our findings revealed reliable actions of the flavonoid hybrids on the maintenance of the free labile iron pool in the neuronal cells, contributing to the beneficial actions of the novel compounds in this paradigm of cell death. Previously, Neitemeier et al. clearly demonstrated in HT22 cells that the prevention of free labile iron in the cytosol by cytosolic iron chelators such as deferoxamine is able to counteract harmful mitochondrial damage induced by glutathione deprivation. Deferoxamine thereby mediated complete preservation of the mitochondrial Δψm, mitochondrial ATP levels, and mitochondrial-derived ROS production [61], confirming that the cytosolic free labile iron pool also has implications for mitochondrial integrity. Beyond the potential impact of the flavonoid–phenolic acid hybrid compounds used in this study, the hybrids also exert direct effects on mitochondria, thereby affecting the detected metabolic shift and, moreover, radical scavenging effects. Therefore, the applied hybrid compounds address multiple targets in the ferroptotic death signaling pathway with high efficacy beyond iron chelation.

The regulation of antioxidant systems is further known to balance the redox state of mitochondria. In some models, quercetin and taxifolin stimulated the expression and activity of endogenous antioxidant enzymes within mitochondria, such as glutathione peroxidase (GPX4), thereby neutralizing ROS and maintaining redox homeostasis. For this hypothesis, we tested GPX4 expression under basal and oxidative stress conditions but did not observe significant regulations by any presence of the tested compounds. However, the Decker Group observed activation of Nrf2—the key regulator of the antioxidant response—and maintained GSH levels in models of oxidative cell death with the compounds UW-MD-189 and -190 [29], giving rise to the activation of other antioxidant and detoxification genes, including those encoded for superoxide dismutase (SOD) and heme oxygenase-1 (HO-1), which were found to be influenced by natural products.

In addition, our data support previous findings demonstrating that mitochondrial integrity and function are essential for preventing ferroptosis. The stabilization of mitochondrial membranes prevents mitochondrial damage and protects against ferroptosis, as disruption of mitochondrial membrane potential can lead to electron leakage and increased ROS production. Flavonoids may support the maintenance of the mitochondrial membrane potential in conditions of cellular stress. Here, we showed that the novel hybrid compounds supported mitochondrial membrane integrity, even after 16 h of oxidative challenge by ferroptosis induction. They also rescued the mitochondrial membrane potential to control levels, which was further reflected by the fully preserved mitochondrial morphology. 

On the other hand, it has been reported that flavonoids can alleviate neurodegeneration via mild mitochondrial uncoupling in *C. elegans* [62,63]. It was suggested that TMRE accumulation in mitochondria depends on mitochondrial activity, and uncoupler or protonophores are able to dissipate the mitochondrial membrane potential, open the mitochondrial membrane for proton flow, and concomitantly weaken the proton gradient. Therefore, electrons can pass down the ETC and ultimately lower mitochondrial ROS production. This effect was observed for several flavonoids, such as quercetin, apigenin, and fisetin, and for the flavone (2-phenyl-4H-1-benzopyran-4one) [62,63]. However, this action was not observed for the UW-MD compounds when investigating the mitochondrial membrane potential with TMRE in hippocampal cells. Interestingly, the quercetin derivate UW-MD-458 diminished mitochondrial membrane potential at concentrations higher than 6.25 µM; this may be due to uncoupling properties such as those reported for quercetin, which has not been detected so far for taxifolin. 

Along with these results and the outstanding properties of UW-MD-458, it was of interest to investigate the mitochondrial Ca^2+^ uptake, as calcium signaling, particularly a Ca^2+^ overload in the mitochondria, is interconnected with mitochondrial membrane potential and can cause a shift toward a membrane hyperpolarization. The regulation of both mitochondrial calcium signaling and mitochondrial membrane potential has an important impact on cellular functions and fate and can result in the formation of the mPTP, release of cytochrome c, and caspase-dependent cell death [45]. However, we did not observe that any of the compounds interfered with caspase-dependent apoptosis induced by staurosporine. As we did not observe a direct effect on the mitochondrial calcium transporter MCU at the protein level, we assume that the maintenance of mitochondrial calcium levels was not mediated by direct effects on this transporter. Most intriguingly, our results indicate that the novel hybrids interfere with ferroptotic signaling by directly targeting mitochondrial respiration. For the compounds UW-MD-457 and UW-MD-458, we showed a strong reduction in complex I activity after only 3 h of incubation, leading to an altered bioenergetic metabolism. Complex I is the first and largest complex in the ETC; here, NADH is oxidized, thereby transferring electrons to ubiquinone (coenzyme Q), and protons are simultaneously pumped across the inner mitochondrial membrane, establishing the proton gradient necessary for ATP production. 

It is generally accepted that there are different classes of complex I inhibitors that have shown distinct effects on mitochondrial ROS production. On the one hand, Class A inhibitors, such as rotenone, piericidin A, or rolliniastatin, were shown to increase ROS production by blocking electron transfer and binding specifically to the N2 iron–sulfur cluster, one of eight iron–sulfur clusters found in complex I and often referred to as the (ubi)quinone-binding site, promoting the forward transport of electrons through the respiratory complexes [64]. Binding to this cluster can lead to reverse electron transport chain (RET) and electron leakage, responsible for the increased ROS production in these conditions [64,65]. However, Class B complex I inhibitors were shown to prevent ROS production. Molecules like capsaicin, stigmatellin, coenzyme Q1, or the novel inhibitor NADH-OH were assumed to bind to another residue, very likely to the NADH binding site of complex I, which does not interfere with the electron flow through the mitochondrial complexes whilst blocking the side reaction of NADH oxidation, leading in turn to harmful ROS production [64,66]. Especially in models of ischemic stroke or ischemia/reperfusion (I/R) injury, based on the increased production of ROS and energy failure caused by changes in mitochondrial metabolism, it was found that RET plays a detrimental role in the deleterious mechanism of stroke pathology [67]. Due to the specific actions of the novel hybrids on complex I and the reliable reduction in ETC-derived ROS (MitoSOX) and the well-preserved mitochondrial membrane potential, the derivatives seem to rely on alternative binding sites related to the ubiquinone-binding site. Despite the considerable progress that has been made recently in understanding the structure, molecular composition, and evolution of complex I and inhibitors, the exact molecular mechanism and alternative binding sites leading to the beneficial protection against mitochondrial ROS remain a matter of speculation [66]. The new compounds therefore have the potential to be a tool for the further elucidation of these interactions.

Besides the direct complex I inhibition over protein binding sites, it has to be taken into account that mitochondrial complex I activity can be influenced through indirect mechanisms. In studies using isolated mitochondria from rat hearts, Ca^2+^ at low micromolar concentrations was able to inhibit complex I through a reduction in NADH-supported electron transport activity. This effect was specific to complex I with no alterations in the activities of other electron transport chain complexes [68,69]. Similar to other class B inhibitors, Ca^2+^ reduced the relative rate of O_2_^•−^ production, consistent with the magnitude of complex I inhibition [69].

Moreover, previous results for metformin, the most widely prescribed medication to treat type II diabetes, demonstrated that the concentrations of metformin required to inhibit complex I are consistently lower in intact cells (micromolar range) than in isolated mitochondria with millimolar concentrations [70], indicating that next to direct complex I-binding sites, metformin is assumed to act on complex I via metabolic regulations such as AMPK activation, blocking gluconeogenesis or altering lipid metabolism. 

Notably, the UW-MD compounds were assumed to act over ANT-1 translocase, also called the mitochondrial ADP/ATP carrier, responsible for the exchange of cytosolic ADP and mitochondrial ATP across the inner mitochondrial membrane. ANT-1 proteins are highly distributed within the mitochondrial membrane and play an important role in the regulation of mitochondrial energy metabolism since the promotion of OXPHOS-derived ATP translocation has an impact on mitochondrial ATP turnover and was further shown to act as an uncoupler of the proton efflux over the ETC [71,72,73]. This is in line with the finding that the hybrid compounds were able to reduce mitochondrial respiration by (indirectly) targeting the ETC complexes and limiting the supply of available ADP for mitochondrial ATP production, resulting in decreased OCR levels. In this context, several post-translational modification sites have been discovered in the ANT-1, which were prone to be modified by nitrosylation, methylation, or acetylation, consequently regulating OXPHOS activity. It is likely that the hybrid compounds also act by binding to these specific residues. Among others, nitrosylation of the Cys57 in ANT-1 was shown to play a role in protecting the heart from ischemia/reperfusion (I/R) injury [74,75]. In addition, ANT-1 is suggested to be a major component of the pro-apoptotic mitochondrial permeability transition pore mPTP [71,76], representing an interesting link to how ANT-1 influences calcium levels and in return is influenced by calcium itself. However, the opening of the mPTP by ANT-1 is dependent on the presence of calcium; increasing calcium concentrations were thereby accompanied by an increase in mPTP activity. Since the hybrid compounds preserved mitochondrial membrane potential and mitochondrial calcium levels, this reinforces the assumption that the compounds are possibly able to preserve mitochondrial integrity by interacting with the ANT-1 exchanger.

However, metabolic adaptations due to complex I inhibition have fundamental implications for cellular function and overall health and lead to compensatory mechanisms, often including an increase in glycolysis to sustain ATP production. Emerging research suggests that controlled complex I inhibition could have potential beneficial effects in the context of neurodegenerative diseases. Recently, it was reported that phytochemicals such as extracts from *Cimicifuga racemosa* mediate protection against oxidative cell damage through a metabolic shift from oxidative phosphorylation to glycolysis in neuronal cells [77,78]. These protective effects against ferroptosis are similar to the ones observed by the UW-MD compounds and also promoted sustained cellular resilience to oxidative stress in vivo [78].

In conclusion, our study shows that the novel flavonoid–phenolic acid hybrids provide pronounced and sustained protection against oxidative damage using multiple antioxidant defense mechanisms. In particular, inhibition of OXPHOS complex I may be a key mechanism of reducing ROS production and mitochondrial disintegration, thereby supporting cellular resilience, which may serve as an effective strategy to prevent age-related oxidative stress. 

## 5. Conclusions

The combined effects of dysregulated cellular Ca^2+^ levels, mitochondrial ROS, elevated iron, and, finally, mitochondrial demise create a complex network of interconnected imbalances in metabolic pathways and redox homeostasis that contribute to the progression of neurodegenerative diseases. The highly potent hybrid compounds presented here may not only have strong pharmacological effects preventing this cascade of harmful events but also have implications for therapeutic strategies, as ferroptosis has been increasingly associated with the pathological processes of cardiovascular diseases, diabetes, and cancer, especially when involving dysregulated mitochondrial metabolism. The development of novel, safe, and effective drugs is needed for the treatment of pathological conditions where oxidative dysregulation and mitochondrial impairment cause the progression of the disease.

## Figures and Tables

**Figure 1 antioxidants-13-00044-f001:**
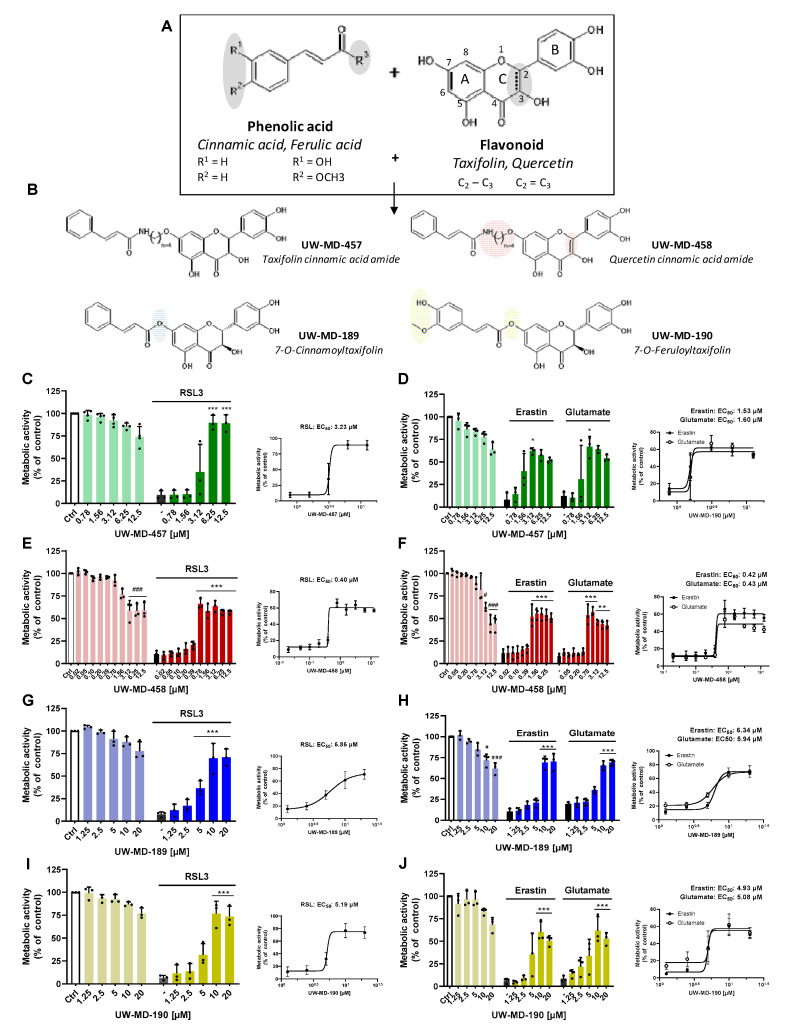
Discovery of novel flavonoid–phenolic acid hybrids and effects in models of ferroptosis. (**A**) Chemical structures of the phenolic acids and flavonoid backbones used for flavonoid–phenolic acid hybridization. Modification sites in the flavonoid and phenolic acid structure of the hybrids are highlighted in gray. (**B**) Novel synthesis of the hybrid compounds UW-MD-457, UW-MD-458, UW-MD-189, and UW-MD-190. Characteristic chemical modifications for the respective compound are highlighted in green, red, blue, or yellow. (**C**–**J**) Dose–response curves of UW-MD compounds in different models of oxidative cell death (0.3 µM RSL3, 0.5 µM erastin, or 7.5 mM glutamate) and respective EC_50_ identification. Results were obtained from cell viability assessment via MTT of three independent experiments with six replicates per sample in HT22 cells after 16 h. Data are shown as mean ± SD; ### *p* < 0.001, # *p* < 0.05 compared to untreated control, *** *p* < 0.001, ** *p* < 0.01, * *p* < 0.05 compared to RSL3, erastin, or glutamate control; ANOVA, Scheffé’s test. Corresponding EC_50_ values were calculated with Prism software 8.2.1.

**Figure 2 antioxidants-13-00044-f002:**
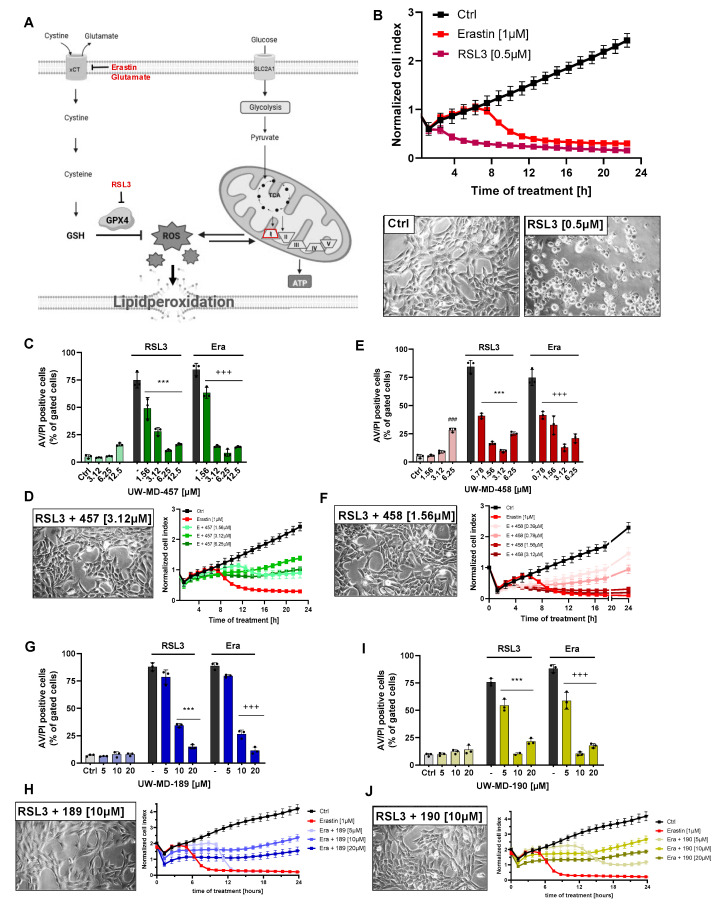
Flavonoid–phenolic acid hybrids mediate strong protection against oxidative cell death. (**A**) Model mechanism of ferroptotic cell death induced with erastin, glutamate, or RSL3. (**B**) Real-time impedance measurements of erastin- and RSL3-induced insult in hippocampal-derived HT22 cells and representative bright-field images of HT22 cells after 16 h treatment with and without RSL3; 20× magnification. (**C**,**E**,**G**,**I**) Cell death detection using Annexin V/PI staining clarifies the dose-dependent protection against erastin-(1 µM) and RSL3 (0.5 µM)-induced ferroptosis meditated by the flavonoid–phenolic acid hybrids according to their EC_50_ concentration after 16 h incubation (n = 3 replicates per condition, percentage of gated cells). ### *p* < 0.001 compared to untreated control, *** *p* < 0.001 compared to RSL3 control, +++ *p* < 0.001 compared to erastin control; ANOVA, Scheffé’s test. (**D**,**F**,**H**,**J**) Representative real-time impedance measurements of erastin and UW-MD co-incubation showed preserved cell proliferation by the hybrid compounds. Individual curves are obtained from 6 replicates per condition and show the mean ± SD; Morphology of RSL3 and UW-MD co-treated HT22 cells was observed under bright field microscope. Representative images are from one of the random fields of view (20× magnification).

**Figure 3 antioxidants-13-00044-f003:**
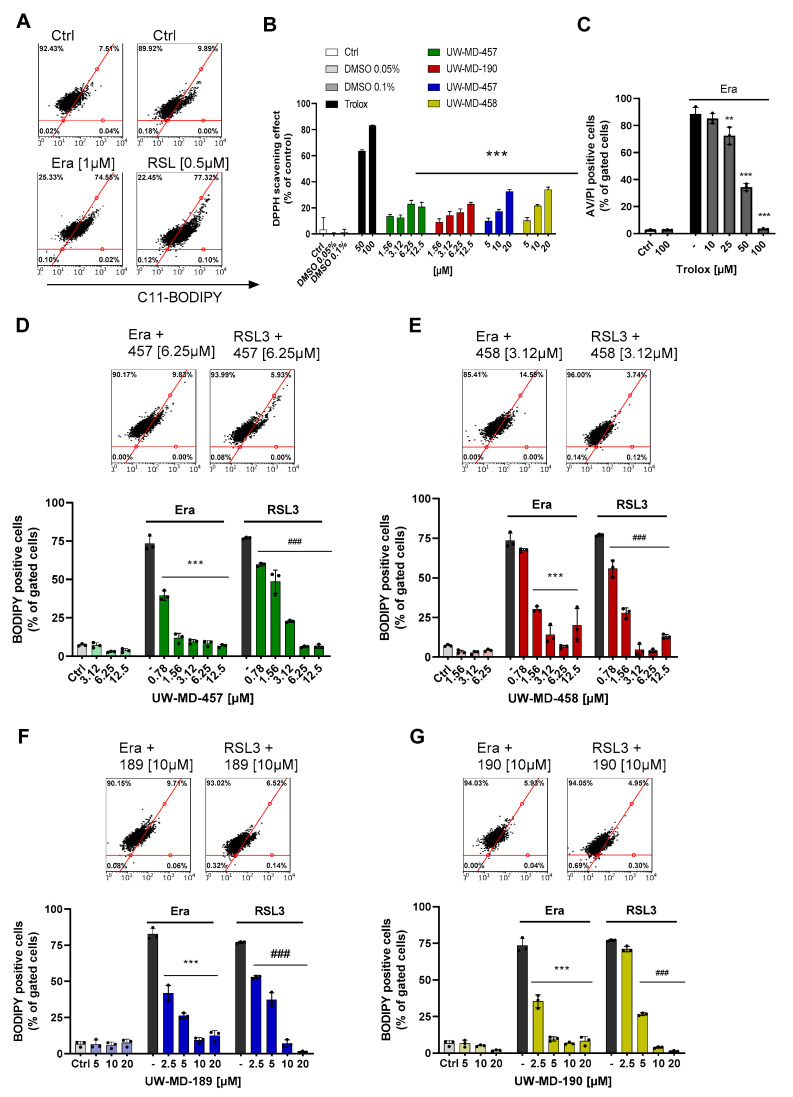
UW-MD compounds prevent lipid peroxidation in models of oxidative stress. (**A**,**D**–**G**) erastin (1 µM) and RSL3 (0.5 µM) led to the formation of lipid peroxidation in HT22 cells after treatment for 8 h. Co-incubation with the flavonoid hybrids reduced ROS-induced C11-Bodipy fluorescence. Data show representative results as mean ± SD from n = 3. *** *p* < 0.001 compared to erastin-treated conditions, ### *p* < 0.001 compared to RSL3 control; ANOVA, Scheffé’s test. (**B**) Antioxidative capacities of the UW-MD hybrids were assessed with the reduction of the stable free radical DPPH, visible by changes in absorbance of DPPH. Radical scavenging activity is shown as mean ± SD, obtained from 6 replicates per condition. *** *p* < 0.001 compared to trolox (50 µM); ANOVA, Scheffé’s test. (**C**) Measurement of cell death with Annexin/PI FACS analysis in response to erastin [1 µM] and trolox after 16 h incubation. Data from n = 3 are shown as mean ± SD; *** *p* < 0.001, ** *p* < 0.01; ANOVA, Scheffé’s test.

**Figure 4 antioxidants-13-00044-f004:**
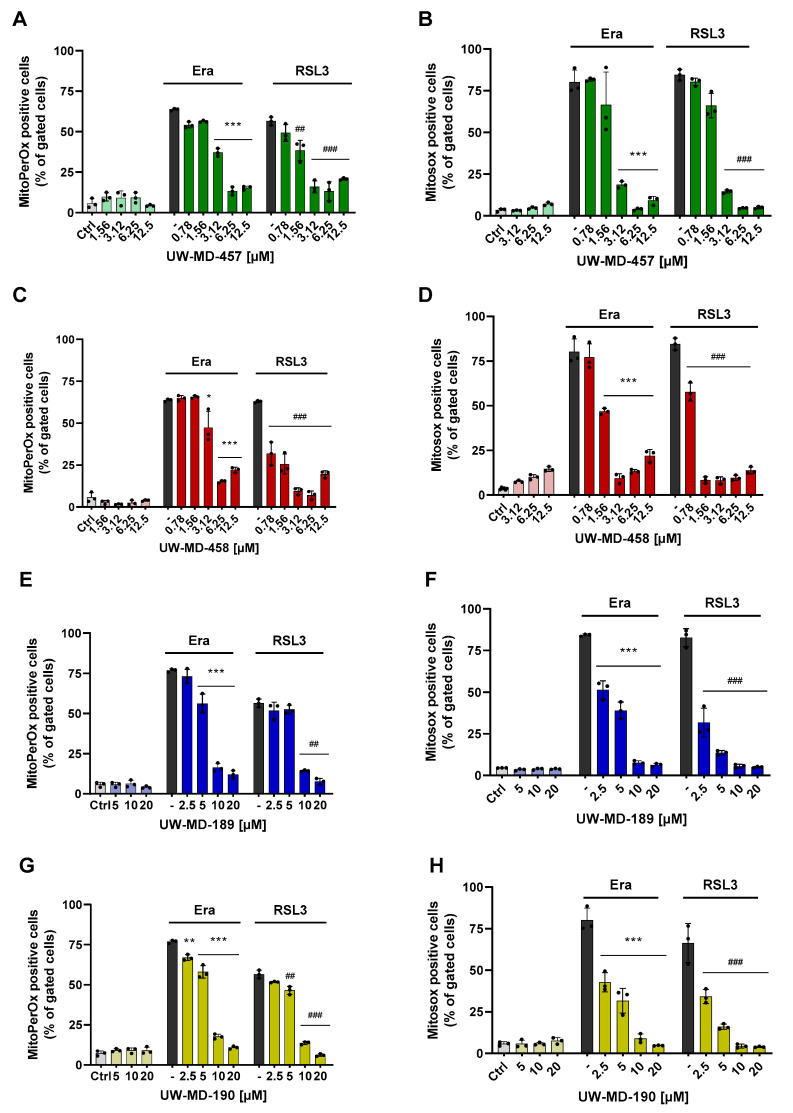
Flavonoid–phenolic acid hybrids reduce mitochondrial-derived ROS formation. (**A**,**C**,**E**,**G**) Mitochondrial lipid peroxidation induced by erastin (1 µM) or RSL3 (0.5 µM) was assessed after 16 h challenge with MitoPerOx staining and subsequent FACS analysis. (**B**,**D**,**F**,**H**) Mitochondrial superoxide (O_2_^•−^) development was detected by MitoSOX red fluorescence after 16 h in HT22 cells. For quantification, 5000 cells per replicate (n = 3) were analyzed and shown as mean ± SD; *** *p* < 0.001, ** *p* < 0.01, * *p* < 0.05 compared to erastin treatment, ### *p* < 0.001, ## *p* < 0.01 compared to RSL3 control; ANOVA, Scheffé’s test.

**Figure 5 antioxidants-13-00044-f005:**
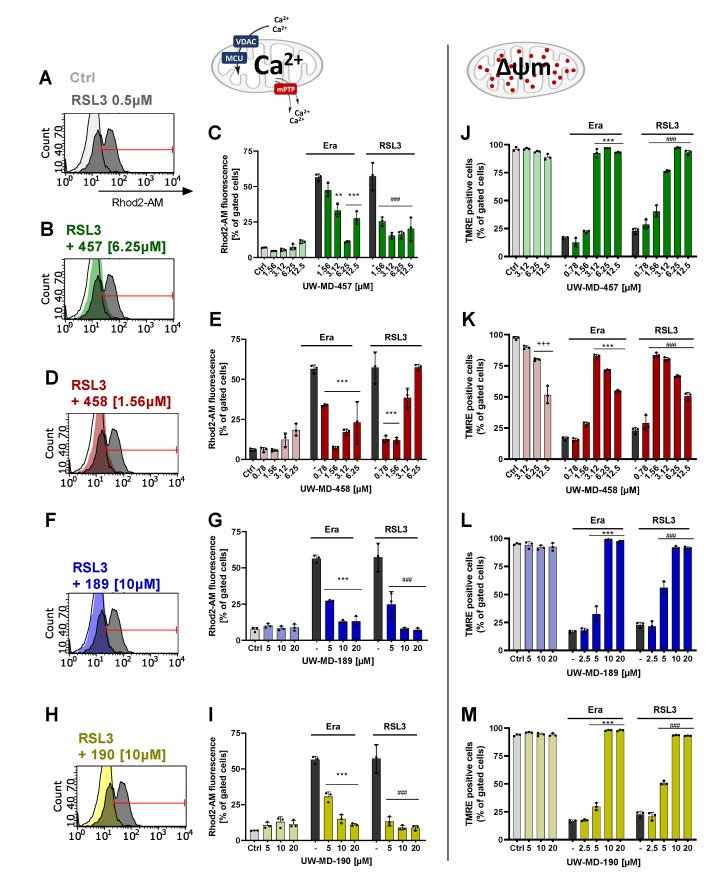
Novel hybrid compounds prevent ferroptotic Ca^2+^ flux into mitochondria, thereby preserving mitochondrial membrane potential. (**A**–**I**) Mitochondrial calcium levels of UW-MD-treated cells were measured via an increase in Rhod-2 AM fluorescence after 16 h in presence or absence of erastin (1 µM) or RSL3 (0.5 µM). (**J**–**M**) Changes in mitochondrial membrane potential were analyzed by the accumulation of a TMRE probe in the mitochondrial membrane after 16 h compound incubation. Each data point represents the mean and standard deviation of n = 3 replicates per condition. +++ *p* < 0.001 compared to untreated control, *** *p* < 0.001, ** *p* < 0.01 compared to erastin treatment, ### *p* < 0.001 compared to RSL3 control; ANOVA, Scheffé’s test.

**Figure 6 antioxidants-13-00044-f006:**
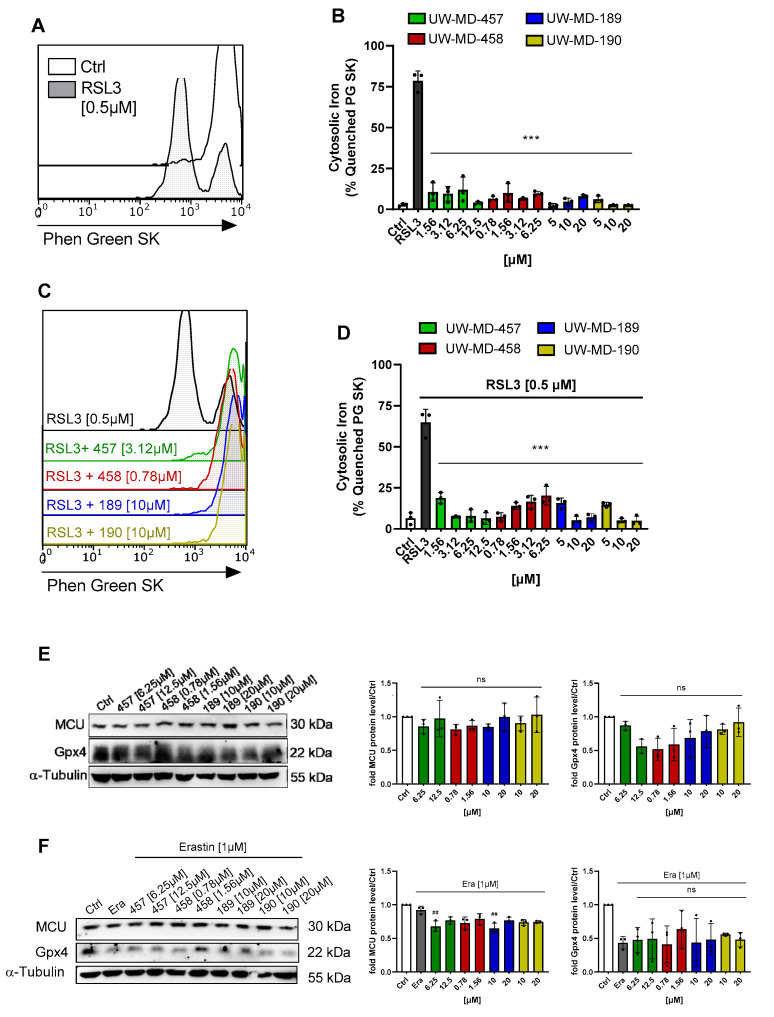
Effects of UW-MD hybrids on intracellular iron content, MCU, and Gpx4 levels. (**A**–**D**) The intracellular labile iron pool upon UW-MD incubation and co-treatment with RSL3 was assessed with PhenGreen SK diacetate staining in HT22 cells after 8 h. Iron accumulation is classified as the proportion of cells showing quenched Phen Green SK fluorescence as shown in the overlaid histograms (**A**,**C**) and quantified on the right (**B**,**D**) from representatives of three replicates per condition. *** *p* < 0.001 compared to RSL3 control; ANOVA, Scheffé’s test. (**E**,**F**) MCU and Gpx4 protein levels were investigated in HT22 cells treated with the single UW-MD compound (**E**) or with a combination of a UW-MD hybrid and 1 µM erastin (**F**) for 8 h and examined by Western blot. ANOVA, Bonferroni’s test; ## *p* < 0.01, ns = not significant compared to erastin-treated control.

**Figure 7 antioxidants-13-00044-f007:**
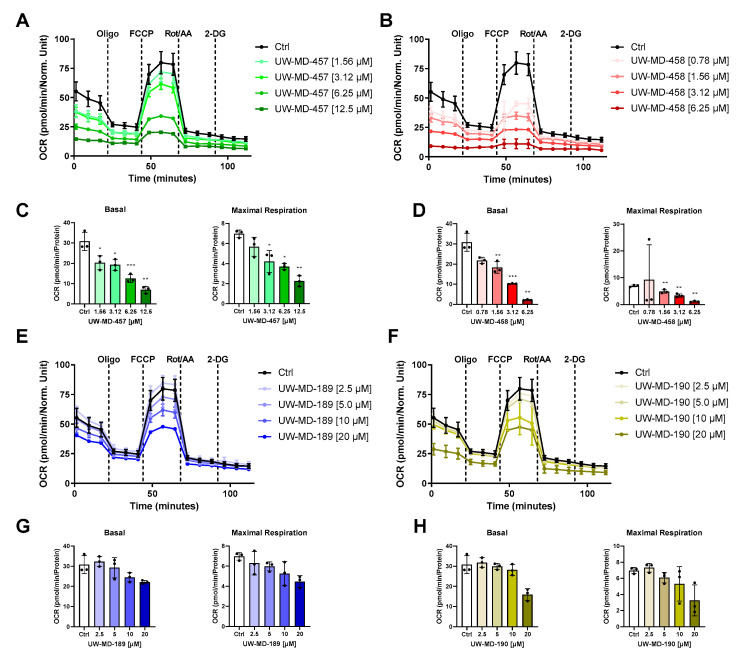
The novel compounds reduce mitochondrial oxidative metabolism. Metabolic profiles of representative oxygen consumption rate (OCR) analysis in HT22 cells treated with UW-MD-457 (**A**), UW-MD-458 (**B**), UW-MD-189 (**E**), or UW-MD-190 (**F**) after 16 h incubation in neuronal HT22 cells (n = 3 replicates per condition). (**C**,**D**,**G**,**H**) Basal respiration was obtained from the average of the first three measurement points before oligomycin application and stated as mean ± SD. Maximal respiration was calculated by the average of the maximal OCR rates after FCCP injection subtracted from the non-mitochondrial respiration apparent by rotenone/antimycin A injection and displayed as mean ± SD. *** *p* < 0.001, ** *p* < 0.01, * *p* < 0.05 compared to control; ANOVA, Scheffé’s test.

**Figure 8 antioxidants-13-00044-f008:**
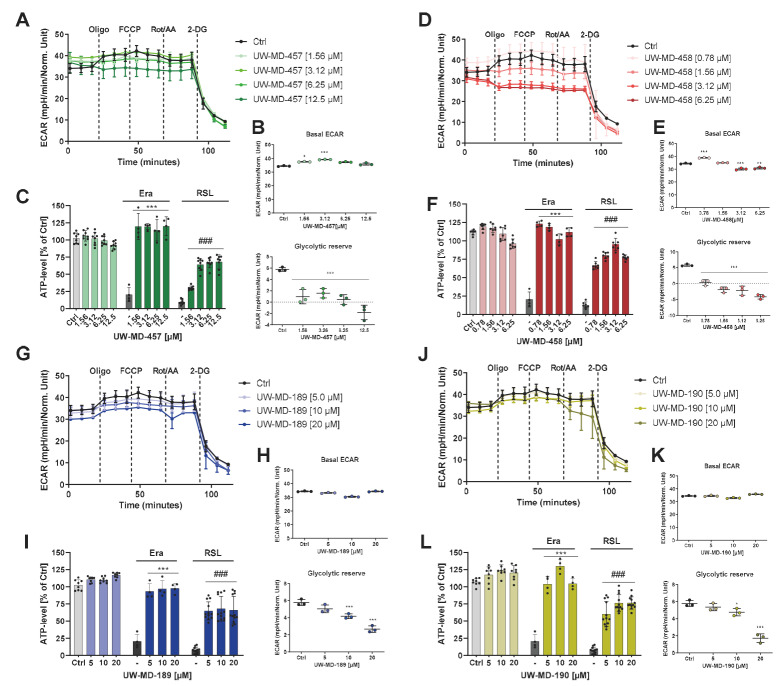
Regulation of the glycolytic profile by UW-MD compounds and effects on cellular ATP levels in HT22 cells. (**A**,**D**,**G**,**J**) The glycolytic activities in the presence of the four compounds were assessed by the extracellular acidification rate (ECAR) and are depicted in one representative glycolytic profile obtained from XF analysis. (**B**,**E**,**H**,**K**) Basal ECAR values were quantified by the average of ECAR values prior to oligomycin injection. Glycolytic reserve was evaluated by the difference in ECAR levels obtained after oligomycin application and basal ECAR measurements of three replicates per condition; data are presented as mean ± SD, *** *p* < 0.001, ** *p* < 0.01, * *p* < 0.5 compared to control, ANOVA, Scheffé’s test. (**C**,**F**,**I**,**L**) Quantification of ATP content in HT22 cells after 8 h of compound treatment with and without the presence of erastin (0.5 µM) and RSL3 (0.3 µM). Each data point represents the mean and standard deviation of n = 3 replicates per condition. ### *p* < 0.001 compared to untreated control, *** *p* < 0.001 compared to erastin treatment, ### *p* < 0.001 compared to RSL3 control, ANOVA, Scheffé’s test.

**Figure 9 antioxidants-13-00044-f009:**
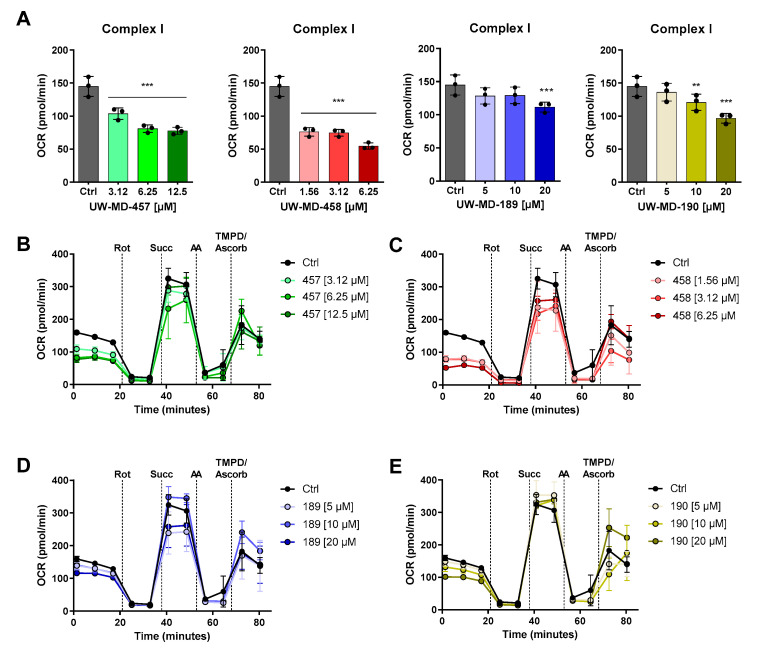
UW-MD hybrids address complex I activity of the mitochondrial electron transport chain. (**A**) Changes in complex I activity were assessed by XF Flux analysis in permeabilized HT22 cells after 3 h pre-incubation with the respective UW-MD compound. Bar charts show the changes in OCR assigned to complex I activity, which was calculated from the received metabolic profiles (**B**–**E**) according to the description in the methods section. Data are represented as mean ± SD, illustrating differences in OCR between compound treatment and control; ANOVA, Scheffé’s test, *** *p* < 0.001, ** *p* < 0.01.

**Figure 10 antioxidants-13-00044-f010:**
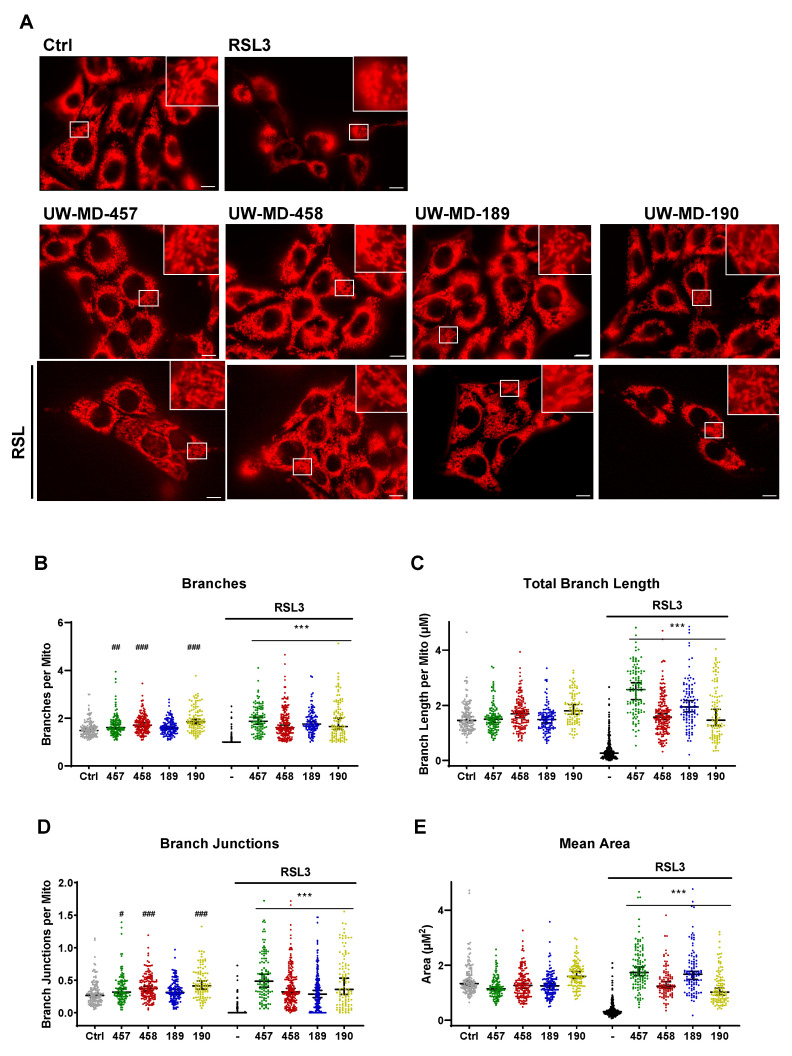
Mitochondrial morphology is preserved by UW-MD compounds. (**A**) Representative images (100× oil objective) of MitoTracker staining after 10 h with and without RSL3 (0.5 µM) challenge. Scale bar: 10 µM. (**B**–**E**) Quantitative comparison of mitochondrial morphology and network connectivity was performed after adaptive thresholding of 50–200 cells per condition with MitoAnalyzer in accordance with the protocol described in the methods section. Data are displayed as the median with 95% CI, ANOVA, Bonferroni’s test, ### *p* < 0.001, ## *p* < 0.01, # *p* < 0.05 compared to untreated control, *** *p* < 0.001 compared to RSL3-treated cells.

**Table 1 antioxidants-13-00044-t001:** EC_50_ values of novel flavonoid–phenolic acid hybrids.

Compound	EC_50_ RSL3	EC_50_ Erastin	EC_50_ Glutamate
UW-MD-189	3.23 µM	1.53 µM	1.50 µM
UW-MD-190	0.40 µM	0.42 µM	0.43 µM
UW-MD-457	5.85 µM	6.34 µM	5.94 µM
UW-MD-458	5.19 µM	4.93 µM	5.08 µM

## Data Availability

Data are contained within the article or Appendix A.

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
