# Peer review of "Flavonoid–Phenolic Acid Hybrids Are Potent Inhibitors of Ferroptosis via Attenuation of Mitochondrial Impairment"

_antioxidants, 2023, doi:10.3390/antiox13010044_

Round 1

Reviewer 1 Report

Comments and Suggestions for Authors

The work uses 4 novel compounds, Flavonoid-Phenolic acid hybrids, on the neuronal cell line HT22 subjected to ferroptosis inducers RSL3 and erastin. The compounds were shown to act as potent inhibitors of ferroptosis, partially by acting as radical scavenging and also, and more importantly, by inhibiting the activity of the mitochondrial complex I and respiration. It is an interesting and well-articulated work.

-           L. 337.  This is figure 2, not 1.

-           I suggest to use always RSL3 in place of RSL In text and figures, to avoid confusion.

-           The origin of Phen Green SK and of DPPH in material and methods should be indicated

- L. 704.  Refer to the publication of the Decker group. 

- The same group showed that one of the compound binds ANT-1  and also SERCA, I was surprised to notice that these interactions that may play a role on the effects of the compounds on the mitochondria and calcium regulation have not been commented in discussion.

- A comparison of the inhibitory activity of the compounds with that of iron chelators would be informative, since they probably bind iron, and this may participate to anti-ferrroptosis.

-  many references include the words "Verfügbar unter:" that is not necessary

Author Response

First of all, we thank the reviewer for careful evaluation of our manuscript and the encouraging detailed comments. We fully agree with the (mainly positive) remarks of the referee and answered all questions raised by the reviewer. We amended presentation and discussion of the data in the revised version of the manuscript accordingly. Please find the detailed answers to the reviewer’s questions below and according changes in the manuscript marked in red font as indicated in detail below.

-           L. 337.  This is figure 2, not 1.

We have fixed the error.

-           I suggest to use always RSL3 in place of RSL in text and figures, to avoid confusion.

Thank you for pointing this out. Since there exist two major types of RSLs (RAS Selective Lethal small molecules), it is important to differentiate these ferroptosis inducers. Type I such as RSL5 require upstream regulators (e.g., VDAC and system XC- to initiate ferroptosis, whereas type II such as RSL3 can trigger ferropotis by inhibiting downstream regulators (e.g., GPX4). In our study, we used the well characterised RSL3 derivative. Therefore, we corrected the notation in all sections and figures accordingly.  

-           The origin of Phen Green SK and of DPPH in material and methods should be indicated

We agree with this and have incorporated the origin of the compounds in the material and methods sections and state now in L. 224: “To measure free labilie Fe2+ in cells, the fluorescent heavy metal indicator Phen Green SK diacetate (Cat#25393, Cayman Chemical, Ann Arbor, MI, USA) with a stock concentration of 1500 µM was used as an indicator.“ We also included in L. 267: “Antioxidant activity of the compounds was determined by DPPH (2,2-diphenyl-1-picrylhydrazyl, (Cat#14805-50, Cayman Chemical, Ann Arbor, MI, USA) assay… .“

-              L. 704.  Refer to the publication of the Decker group. 

In the revised version of the manuscript L. 714, we included the reference from Gunesch et al., 7-O-Esters of taxifolin with pronounced and overadditive effects in neuroprotection, anti-neuroinflammation, and amelioration of short-term memory impairment in vivo. Redox Biology 29 (2020), S. 101378.

-              The same group showed that one of the compound binds ANT-1 and also SERCA, I was surprised to notice that these interactions that may play a role on the effects of the compounds on the mitochondria and calcium regulation have not been commented in discussion.

Thank you for bringing attention to the previous findings regarding the compound’s binding to both ANT-1 and SERCA. In the revised discussion section, we have paid attention to this topic and discussed the potential interaction between ANT-1 and the hybrids and added a corresponding remark in the discussion L. 798: “Notably, the UW-MD compounds were assumed to act over ANT-1 translocase, also called the mitochondrial ADP/ATP carrier, responsible for the exchange of cytosolic ADP and mitochondrial ATP across the inner mitochondrial membrane. ANT-1 proteins are highly distributed within the mitochondrial membrane and play an important role in the regulation of the mitochondrial energy metabolism, since the promotion of the OXPHOS-derived ATP translocation has an impact on mitochondrial ATP turnover and was further shown to act as an uncoupler of the proton efflux over the ETC (75, 77, 78). This is in line with the finding that the hybrid compounds were able to reduce mitochondrial respiration by (indirectly) targeting the ETC complexes and limiting the supply of available ADP for mitochondrial ATP production, resulting in decreased OCR levels. In this context, there are several post-translational modification sites discovered in the ANT-1, which were prone to be modified by nitrosylation, methylation or acetylation, consequently regulating the OXPHOS activity. It is likely that the hybrid compounds also act by binding to these specific residues. Among others, nitrosylation of the Cys57 in ANT-1 was shown to play a role in protecting the heart from ischemia/reperfusion (I/R) injury (79, 80). In addition, ANT-1 is suggested to be a major component of the pro-apoptotic mitochondrial permeability transition pore mPTP (75, 76), representing an interesting link how ANT-1 is influencing calcium level and in return is influenced by calcium itself. However, opening of the mPTP by ANT-1 is dependent on the presence of calcium, increasing calcium concentrations were thereby accompanied with an increase of the mPTP activity. Since the hybrid compounds preserved mitochondrial membrane potential and mitochondrial calcium levels, this reinforces the assumption that the compounds possibly are able to preserve the mitochondrial integrity by interacting with the ANT-1 exchanger.

In addition, previous findings suggested effects of the flavonoid-phenolic acid hybrids on SERCA which may affect ER calcium homeostasis and, indirectly, ER-mitochondrial calcium transfer. Whether effects of the hybrid compounds on SERCA were involved in the observed maintenance of mitochondrial Ca2+ homeostasis under conditions of ferroptosis induction or whether this was attributed to direct effects of the hybrid compounds on mitochondrial integrity and metabolic regulation, is matter of further investigation.

-              A comparison of the inhibitory activity of the compounds with that of iron chelators would be informative, since they probably bind iron, and this may participate to anti-ferrroptosis.

We absolutely agree with the reviewer, that comparison with structurally divergent iron chelators would be of interest. We actually have previously used such an assay for ferulic acid-containing hybrids (M. Scheiner et al. Selective pseudo-irreversible butyrylcholinesterase inhibitors transferring antioxidant moieties to the enzyme show pronounced neuroprotective efficacy in vitro and in vivo in an Alzheimer’s disease mouse model. J. Med. Chem. 2021, 64, 9302-9320). This assay was based on ferrozine and pyrocatechol complex formation of Fe2+ and Cu2+. In our hands, this assay was only yielding quite summative potency values regarding the formation of chelates, which turned out quite hard to quantify accurately. We therefore did not apply this assay in this manuscript.

Instead we referred in L. 699 to our previous work, where we investigated the effects of the cytosolic iron chelator deferoxamine in models of oxidative stress induced by the XC- inhibitor glutamate in hippocampal HT22 cells. Indeed, these results showed a potent protection of mitochondrial characteristics by deferoxamine, which were impaired under oxidative stress conditions. Since the four compounds were able to prevent cytosolic free labile iron accumulation, as detected by quenched Phen Green SK diacetate fluorescence, this gives rise for the assumption that prevention of free labile iron accumulation contribute to the beneficial effects mediated by our tested compounds. Following the reviewer’s notion, we included the remark: “Previously, Neitemeier et al. clearly demonstrated in HT22 cells that the prevention of free labile iron in the cytosol by cytosolic iron chelators such as deferoxamine, is able to counteract the harmful mitochondrial damage, induced by glutathione deprivation. Deferoxamine was thereby mediating complete preservation of the mitochondrial Δψm, the mitochondrial ATP levels and mitochondrial-derived ROS production (81), confirming that the cytosolic free labile iron pool also has implications for mitochondrial integrity. Beyond potential impact of the flavonoid-phenolic acid hybrid compounds used in this study, the hybrids also exert direct effects on mitochondria thereby affecting the detected metabolic shift and, moreover, radical scavenging effects. Therefore, the applied hybrid compounds address multiple targets in the ferroptotic death signaling pathway with high efficacy beyond iron chelation.”

-              many references include the words "Verfügbar unter:" that is not necessary

We thank the reviewer for careful evaluation and detecting this error. We corrected all references accordingly.

Reviewer 2 Report

Comments and Suggestions for Authors

The manuscript provides new ideas for the treatment of neurodegenerative diseases based on flavonoid-phenolic acid mixtures as a way to reduce oxidative stress and protect mitochondria. The manuscript is well structured with convincing details, and the results and discussion provide a solid conclusion to this work. However, there are still some problems with the manuscript, and minor revisions are suggested to make it more understandable and reasonable.

1. The article has several images, such as Figure 3, where the splicing aspect can be optimized, as well as the color of some of the data figures can be improved.

2. More attention should be paid to vocabulary and writing format. For example, inactivated is misspelled in Line 149. Some singular and plural, article and tense usages need to be corrected. It is recommended that revisions be made by native English speakers to make this manuscript more understandable and scientific.

3. The article has a lot of experimental data that could be reflected in the conclusion to make it richer.

4. I would like to know what advantages it has over other methods of treating neurodegenerative diseases.

Comments on the Quality of English Language

More attention should be paid to vocabulary and writing format. For example, inactivated is misspelled in Line 149. Some singular and plural, article and tense usages need to be corrected. It is recommended that revisions be made by native English speakers to make this manuscript more understandable and scientific.

Author Response

We thank the reviewer for the time spent carefully reviewing the manuscript, and are grateful for the insightful comments on science and presentation of the material. We answered all questions raised by the reviewer and accordingly amended presentation and discussion of the data in the revised version of the manuscript. Please find the detailed answers to the reviewer’s question below and according changes in the manuscript marked in red font as indicated.

  • The article has several images, such as Figure 3, where the splicing aspect can be optimized, as well as the color of some of the data figures can be improved.

We agree that optimizing the splicing aspect in Figure 3 and improving the color scheme in certain data figures could enhance the overall visual clarity of our work. We thoroughly addressed these aspects by rearranging figure 3 and by synchronizing the color layout of the individual figures 2, 3, 4, 5, 8 and S4 ensuring that all illustrations display a uniform color design.

  • More attention should be paid to vocabulary and writing format. For example, inactivated is misspelled in Line 149. Some singular and plural, article and tense usages need to be corrected. It is recommended that revisions be made by native English speakers to make this manuscript more understandable and scientific.

We acknowledge the oversight in the misspelling of "inactivated" in Line 149 and removed the error. Additionally, the manuscript was carefully revised for grammar and spelling by a native English speaker. With the revised and edited version of the manuscript, we are now ensuring clarity and scientific accuracy. The manuscript was carefully revised to enhance the readability and adherence to scientific standards.

  • The article has a lot of experimental data that could be reflected in the conclusion to make it richer.

The remark is well taken and, of course, a more comprehensive inclusion of our data could contribute to a more nuanced interpretation of our findings. In the revised version, we highlighted the interconnection and implications of our investigations and clarified the conclusion in L. 842: “The combined effects of dysregulated cellular Ca2+ levels, mitochondrial ROS, elevated iron, and, finally, mitochondrial demise create a complex network of interconnected imbalances in metabolic pathways and redox-homeostasis that contribute to progression of neurodegenerative diseases. The highly potent hybrid compounds presented here may not only have strong pharmacological effects preventing this cascade of harmful events, but also have implications for… .“

  • I would like to know what advantages it has over other methods of treating neurodegenerative diseases.

We thank the reviewer for pointing out this important issue. In contrast to medical herbal formulations, which are extremely complex mixtures of many pharmacologically active compounds accompanied with an extensive network of pharmacological effects, our novel developed compounds are well characterized in their chemical structure, consequently, effectiveness and safety can be clearly attributed to the defined substances. With modern analytical techniques now available, based on network pharmacology, molecular docking and in vitro cell-based investigations, it is possible to draw a precise profile of the operational pharmacological networks of the hybrid compounds, due to the exact structural definition of the hybrids, which is not given in comparison to common herbal preparations.

In addition, the natural-derived flavonoid-phenolic acid hybrids may have fewer side effects compared to other synthetic drugs. This is especially relevant when considering long-term treatments for chronic conditions like neurodegenerative diseases and offer potential for prevention or delaying the onset of neurodegeneration such as observed in Alzheimer’s disease. These so far unknown effects are matter of ongoing extensive studies and beyond the scope of this study.

  • More attention should be paid to vocabulary and writing format. For example, inactivated is misspelled in Line 149. Some singular and plural, article and tense usages need to be corrected. It is recommended that revisions be made by native English speakers to make this manuscript more understandable and scientific.

We acknowledge the oversight in the misspelling of "inactivated" in Line 149 and removed the error. Additionally, the manuscript was carefully revised for grammar and spelling by a native English speaker. With the revised and edited version of the manuscript, we are now ensuring clarity and scientific accuracy. The manuscript was carefully revised to enhance the readability and adherence to scientific standards.